# Anti-Angiogenic Activity of Drugs in Multiple Myeloma

**DOI:** 10.3390/cancers15071990

**Published:** 2023-03-27

**Authors:** Ilaria Saltarella, Concetta Altamura, Carmen Campanale, Paola Laghetti, Angelo Vacca, Maria Antonia Frassanito, Jean-François Desaphy

**Affiliations:** 1Department of Precision and Regenerative Medicine and Ionian Area, Section of Pharmacology, School of Medicine, University of Bari Aldo Moro, 70124 Bari, Italy; 2Department of Precision and Regenerative Medicine and Ionian Area, Section of Internal Medicine, School of Medicine, University of Bari Aldo Moro, 70124 Bari, Italy; 3Department of Precision and Regenerative Medicine and Ionian Area, Section of Clinical Pathology, School of Medicine, University of Bari Aldo Moro, 70124 Bari, Italy

**Keywords:** angiogenesis, anti-angiogenic drugs, pharmacology, multiple myeloma

## Abstract

**Simple Summary:**

The “angiogenic switch” contributes to the transition from monoclonal gammopathy of undetermined significance (MGUS) to multiple myeloma (MM), sustaining disease progression and drug resistance. In this review, we describe the anti-angiogenic activity of drugs employed in clinical settings for the management of MM patients providing important information on their pharmacological profile and on the molecular mechanisms responsible for their direct and indirect anti-angiogenic effect.

**Abstract:**

Angiogenesis represents a pivotal hallmark of multiple myeloma (MM) that correlates to patients’ prognosis, overall survival, and drug resistance. Hence, several anti-angiogenic drugs that directly target angiogenic cytokines (i.e., monoclonal antibodies, recombinant molecules) or their cognate receptors (i.e., tyrosine kinase inhibitors) have been developed. Additionally, many standard antimyeloma drugs currently used in clinical practice (i.e., immunomodulatory drugs, bisphosphonates, proteasome inhibitors, alkylating agents, glucocorticoids) show anti-angiogenic effects further supporting the importance of inhibiting angiogenesis from potentiating the antimyeloma activity. Here, we review the most important anti-angiogenic therapies used for the management of MM patients with a particular focus on their pharmacological profile and on their anti-angiogenic effect in vitro and in vivo. Despite the promising perspective, the direct targeting of angiogenic cytokines/receptors did not show a great efficacy in MM patients, suggesting the need to a deeper knowledge of the BM angiogenic niche for the design of novel multi-targeting anti-angiogenic therapies.

## 1. Introduction

Multiple myeloma (MM) is a hematological neoplasia that involves monoclonal malignant plasma cells (MM cells), which accumulate in the bone marrow (BM) and release high levels of monoclonal immunoglobulins leading to the pathological manifestations, i.e., bone disease, anemia, renal impairment, hypercalcemia, and hyperuricemia [1]. Usually, MM is preceded by two preneoplastic stages, namely monoclonal gammopathy of undetermined significance (MGUS) and smoldering myeloma (SMM), with an increased risk of progressing to full-blown MM [2].

Several studies have shown that the transition from MGUS to MM is driven by substantial modifications of BM stromal cells (BMSCs) that, together with tumor cells, contribute to shape a tumor niche where the malignant clone proliferates and expands [3,4]. A hallmark of this process is the “angiogenic switch” characterized by the formation of new blood vessels. Enhanced angiogenesis, together with other factors (i.e., cytokines, extracellular vesicles, immune escape, ncRNAs), fosters MM progression and drug resistance [5]. Vacca and collaborators [6] first observed the increased microvessel density (MVD) in patients with active MM compared to remission phase MM and MGUS ones, suggesting that BM angiogenesis correlates with the disease stage [6]. Many other studies have demonstrated a significant correlation between high levels of circulating angiogenic cytokines and MM patient’s prognosis and/or response to therapy indicating that BM MVD may represent an index of progressive disease and shorter progression-free survival [7,8,9].

Based on the pivotal role of angiogenesis in MM progression and its impact on patients’ prognosis, anti-angiogenesis therapy represents an attractive tool for the treatment of MM patients [10,11]. Furthermore, many antimyeloma drugs have shown secondary anti-angiogenic properties in vitro and in vivo, suggesting a promising potential for angiogenesis targeting. In this review, we describe the most important drugs with a direct and indirect anti-angiogenic effects used in MM settings.

## 2. Angiogenesis and Vasculogenesis in Multiple Myeloma

Aberrant angiogenesis is a key hallmark of MM progression. Both BMSCs and MM cells contribute to shape the BM angiogenic niche leading to the sprouting of pre-existing blood vessels, i.e., angiogenesis, and/or to a de novo vessel formation by recruiting CD34^+^ endothelial progenitor cells (EPCs), i.e., vasculogenesis [6,12].

During the transition from the avascular to the vascular phase, the activation of oncogenes such as c-myc, c-fos, c-jun, and Jun-B induces MM cells to secrete high amounts of pro-angiogenic cytokines, including vascular endothelial growth factor (VEGF), fibroblast growth factor 2 (FGF-2), hepatocyte growth factor (HGF), angiopoietin-1, and insulin-like growth factor 1 (IGF-1) [13,14,15]. In turn, these cytokines act on BMSCs and on MM cells as well. For instance, VEGF released by MM cells binds to VEGF receptor 2 (VEGFR2) on endothelial cells (ECs) of MM patients (MMECs) and to VEGFR1 on BMSCs, triggering their proliferation, chemotaxis as well as the release of other angiogenic cytokines sustaining the VEGF-paracrine loop [16]. On the other side, VEGF also acts in an autocrine manner on MM cells themselves via VEGFR1, enhancing their survival, proliferation, and further VEGF release through the activation of the ERK pathway [17]. Stimulation of the VEGF/VEGFR signaling also induces the secretion of IL-6 by BMSCs that, in turn, sustains MM cell growth and survival, further supporting MM pathogenesis [18]. Similarly, Ferrucci and collaborators [19] demonstrated the existence of an autocrine HGF/cMET loop in MMECs, which regulates several angiogenic activities [19] and induces HGF to release that sustains MM cells survival in a paracrine fashion [20]. Accordingly, dysregulation of the cMET pathway represents a poor prognostic factor for patients [21]. FGF-2 is another key factor that significantly increases the BM sera of patients [22]. MM cells and BMSCs produce high levels of FGF-2 that stimulate MMEC proliferation, survival, migration, and angiogenesis [6].

In addition to these cytokines, other growth factors, such as chemokines and enzymes (i.e., angiopoietin-1, angiopoietin-2, IGF-1, MMP-2, MMP-9), promote BM angiogenesis, MMECs migration, and chemotaxis and are considered pivotal players during the MM “angiogenic switch” [5,23,24,25,26].

The high levels of angiogenic factors in the BM milieu stimulate the expression of their cognate receptors on ECs. Accordingly, MMECs express higher levels of angiogenic receptors including VEGFR, HGFR, and FGFR2, compared to ECs from MGUS patients (MGECs), suggesting their higher responsiveness to angiogenic factors [5,6,27].

Due to this pro-angiogenic environment, MMECs acquire an activated phenotype characterized by resistance to apoptosis, increased proliferation rate, and enhanced angiogenic activities, i.e., adhesion to the extracellular matrix (ECM), proliferation, migration, and vascular development [5,26]. Nevertheless, blood vessels from tumor sites have a discontinuous wall characterized by structural and functional abnormalities that generate a hypoxic area leading to BM acidosis, HIF-1α activation, and further VEGF release, which foster aberrant angiogenesis in MM patients [27,28].

Other cells of the BM milieu sustain this complex network, including fibroblasts via cytokine release [29], osteoblasts and osteoclasts via osteoprotegerin (OPG) and the RANK pathway [30], and immune cells [31]. In particular, macrophages and mast cells from MM patients acquire an ECs-like phenotype, express ECs markers, and actively participate in the formation of the angiogenic wall, known as vasculogenesis mimicry [32,33].

Finally, recent studies have described the involvement of long non-coding RNA [34], microRNAs (i.e., miR-135, miR-15a, and miR-16) [35,36], and extracellular vesicles that deliver cytokines/growth factors in promoting BM angiogenesis [37].

The most important mechanisms involved in MM angiogenesis are depicted and summarized in Figure 1.

## 3. Anti-Angiogenic Drugs in MM

Over time, several drugs directly targeting angiogenic cytokines and/or pathways (i.e., monoclonal antibodies (mAbs), bispecific molecules, tyrosine kinase inhibitors (TKIs)) or indirectly affecting angiogenesis (i.e., immunomodulatory drugs (IMiDs), proteasome inhibitors, alkylating agents, bisphosphonates, glucocorticoids) have been developed and used in the MM setting (Figure 2).

### 3.1. Anti-Angiogenic Drugs in Multiple Myeloma

#### 3.1.1. Monoclonal Antibodies

As the VEGF/VEGFR2 axis represents one of the most important pathways in tumor angiogenesis, monoclonal antibodies (mAbs) targeting VEGF or its cognate receptors have been developed (Figure 2a).

Bevacizumab is a humanized anti-VEGF-A IgG1 mAbs, which prevents the binding of VEGF to its cell surface receptors [38]. This inhibition leads to a reduction in microvascular growth of tumor vessels, limiting blood perfusion and tissue interstitial pressure, thus increasing vascular permeability. These modifications result in a rise in chemotherapy delivery and ECs death through apoptosis [39]. Bevacizumab may disrupt VEGF/VEGFR autocrine and paracrine loops [16,17], counteracting BM angiogenesis and MM cell survival. Attar-Schneider and collaborators [40] showed that VEGF inhibition through bevacizumab reduces cell viability in a dose-dependent manner and halts MM cells in the G1 phase of the cell cycle by reducing VEGFR1 activation. Furthermore, inhibition of the VEGF/VEGFR1 axis affects mTOR, Akt, and STAT-3 phosphorylation, c-Myc levels, and the activity of the translation initiation factor, eIF4E, suggesting that bevacizumab is active on different MM cells pathways including proteostasis [40].

Bevacizumab is currently approved by FDA as first-line therapy for the treatment of metastatic colorectal cancer, metastatic non-squamous non-small cell lung cancer, metastatic renal cell carcinoma, and cervical/ovarian cancer in different combined regimens [41]. In MM, different clinical trials of bevacizumab alone (NCT00482495) or combination with other antimyeloma drugs, including bortezomib (NCT00473590, NCT00464178), thalidomide (NCT00022607), lenalidomide and dexamethasone (NCT00410605), have been investigated and/or are under investigation for the treatment of relapsed or refractory patients. Common adverse events are hypertension, fatigue, neutropenia, anemia, and increased infections of the upper respiratory tract [42,43].

Endpoint analysis of patients treated with combo bevacizumab-bortezomib or bevacizumab-thalidomide did not show a great improvement of patient’s outcome [42,43]. In the AMBER trial, the overall response rate in bevacizumab-bortezomib-treated patients was only slightly higher compared to patients treated with bortezomib as single agent (51% vs. 43.4%) [42]. Similarly, among the 14 patients treated with bevacizumab and/or thalidomide in the California Cancer Consortium trial, two patients had a partial response and three attained a stable disease achieving overlapping results to the thalidomide arm [43]. Immunohistochemistry analysis of VEGF and VEGFR1/2 expression on MM cells from patients’ BM biopsies suggests a correlation between their expression and response to therapy [43]. These unfavorable results could be related to the complexity of tumor angiogenesis, which involves not only the VEGF axis but also alternative cytokines and intracellular pathways [44,45] as well as miRNAs [35,36] and extracellular vesicles [37], highlighting that anti-angiogenic treatment should entail the simultaneous targeting of multiple intracellular pathways.

More recently, other anti-VEGF (i.e., the mAb fragment ranibizumab) [46] and anti-VEGFR (i.e., ramucirumab, tanibirumab) [47,48] targeting antibodies have been developed. Nevertheless, to date, these mAbs have not yet been used in clinical practice.

#### 3.1.2. Bispecific Molecules and Recombinant Molecules Targeting Angiogenic Cytokines

Dysregulation of cytokines and growth factors is a key feature that drives aberrant angiogenesis. As previously described, upregulation of VEGF, HGF, FGF-2 and other pro-angiogenic factors are involved in a reduced response to therapy and poor prognosis [5,6,7,8,9]. Accordingly, different drugs able to neutralize these angiogenic cytokines have been developed.

MP0250 is a recently developed multi-domain ankyrin repeat protein (Darpin) composed by an anti-VEGF-A, an anti-HGF, and two anti-human serum albumin (HSA) modules in a single linear protein [48]. The anti-VEGF and anti-HGF motifs bind with high affinity and neutralize both VEGF-A and HGF, whereas the anti-HSA modules enhance MP0250 plasma half-life and tumor penetration potentiating the drug’s pharmacokinetic ([49], NCT03136653) (Figure 2b). Rao and collaborators [50] demonstrated that the MP0250 molecule reduces the phosphorylation of VEGFR2 and cMet receptors on MMECs, inhibiting the activation of downstream pathways. Inhibition of these pathways restrains both angiogenesis in vitro and the release of several angiogenic cytokines. MP0250 also affects angiogenesis in vivo by reducing the formation of new blood vessels in the choriollantoic membrane assay (CAM) as well as in the 5T33MM mouse model [50]. Interestingly, MP0250 gives a synergistic anti-angiogenic and antimyeloma effect in conjunction with bortezomib [50].

MP0250 has been tested in a phase II clinical trial in combination with bortezomib and dexamethasone in relapsed/refractory MM patients previously treated with proteasome inhibitors and IMiDs. Pharmacokinetics data showed that MP0250 has an 11 days half-life. Accumulation of MP0250-HGF complexes implies a stable MP0250/HGF binding with all circulating HGF trapped. Preliminary results showed that hypertension, thrombocytopenia, proteinuria, and increase in transient liver enzymes are the most frequent adverse events [51].

Another strategy to inhibit angiogenic cytokines in the BM microenvironment consists of the use of the recombinant fusion protein, aflibercept [52], also known as VEGF-Trap. It recognizes and neutralizes VEGF-A, VEGF-B, and placental growth factor (PIGF), hindering VEGF-induced angiogenesis [51] (Figure 2b). Currently, aflibercept is in phase II for the treatment of stage II/III MM patients who relapsed or were refractory to previous treatments. Data on the primary and secondary endpoints of this trial are not yet available. Based on clinical studies on solid tumors (i.e., ovarian cancer, metastatic colorectal cancer, non-small cell lung adenocarcinoma), aflibercept treatment should induce low-grade and reversible toxicities, including hypertension and proteinuria, and should stimulate an anti-angiogenic and anti-tumor response, thus reducing tumor growth ([53], NCT00437034).

#### 3.1.3. Tyrosine Kinase Inhibitors

As multiple signaling cascades involved in cell survival, proliferation, and angiogenesis strongly rely on the activation of tyrosine kinase receptors [5,6], inhibition of these pathways has been considered a valuable strategy in cancer. Indeed, several tyrosin kinases inhibitors (TKIs), i.e., sorafenib, vandetanib, pazonib, and dasatinib, have been developed and employed in solid and hematological malignances, including MM (Figure 2c).

Sorafenib is a small orally available multi-kinase inhibitor that simultaneously targets Raf, VEGF, and PDGF receptors [54]. It is a lipophilic molecule with low-solubility and high permeability, which is rapidly absorbed by the gastrointestinal tract and metabolized in the liver through the CYP3A4 and UGT1A9 pathways leading to still active metabolites [55]. Thus, genomic inter-individual differences in cytochromes P450 enzymes cause differences in sorafenib pharmacokinetics and efficacy. The majority of sorafenib is eliminated in the feces and only a small, metabolized part in the urine [55].

Sorafenib induces apoptosis of MM cell lines and primary CD138^+^ MM cells via caspase-dependent and caspase-independent mechanisms in the presence of BMSCs, thus abrogating the protective effect of the BM milieu. Sorafenib has shown in vitro synergistic cytotoxic effects with autophagy inhibitors (i.e., chloroquine), with the mTORC1 inhibitor, rapamycin, and with the commonly used antimyeloma drugs (dexamethasone and bortezomib) [56,57]. Furthermore, sorafenib treatment has shown an anti-angiogenic effect in vitro by reducing tube formation of marrow stromal cells alone or co-cultured with MM cells [57]. Finally, in vivo studies using the 5T33MM mouse model showed that sorafenib improves overall survival and significantly decreases tumor burden and MVD in the BM of treated mice compared to controls [56]. To date, sorafenib has been tested in different trials alone or in combination with other antimyeloma drugs, i.e., bortezomib, lenalidomide, and dexamethasone. Phase II trials showed few adverse events of mild or moderate grade (i.e., fatigue, pain, nausea, rash, hand-foot syndrome) and only one grade 4 toxicity due to cardiac infarction ([58], NCT00253578). Despite the promising preclinical results, a phase II trial of sorafenib alone showed only partial effect [58], and other trials testing combo sorafenib/antimyeloma drugs have been early closed or did not move to phase II due to lack of efficacy (NCT00536575, NCT00687674).

Vandetanib, a TKI-targeting VEGFR2, epidermal growth factor receptor (EGFR), rearranged during transfection (RET) tyrosine kinases, Bruton’s tyrosine kinase, and other cellular kinases (i.e., EPH receptor and Src), significantly reduces the activation of the VEGFR2 pathway in human umbilical vein endothelial cells (HUVEC) and affects their VEGF-dependent proliferation [59,60]. Oral administration of vandetanib to different xenograft mouse models (lung, prostate, breast, ovarian, colon, or vulval) significantly decreased tumor growth and tumor-induced neovascularization [60]. Analysis of pharmacokinetics data from a phase II study on relapsed MM patients showed that the oral administration of vandetanib (100 mg/daily) displays a large volume of distribution, slow absorption, and long half-life [60]. The most common treatment-related adverse events are gastrointestinal symptoms, skin rash, hypertension, and asymptomatic QTc prolongation [61,62]. Unfortunately, despite it achieving plasma levels able to inhibit VEGF signaling, no MM patients responded to therapy, and 50% of them showed progressive disease, leading to trial interruption [62].

Pazopanib (GW786034B) is a small molecule able to inhibit VEGFR in MM cells and in MMECs, abrogating VEGF autocrine and paracrine loops. It reduces MM cell growth, survival, and migration, and disrupts MM cells/MMECs interactions. In vivo studies showed that VEGFR inhibition reduces tumor growth and angiogenesis and enhances mice survival with promising synergistic effects in combination with melphalan, bortezomib and IMiDs [63]. Nevertheless, pazopanib did not show any therapeutic effect in MM patients enrolled in a phase II trial despite the suitable pharmacokinetic and evidence of pharmacodynamic effects on VEGFR inhibition such as depigmentation and hypertension [64].

Finally, dasatinib is a TKI originally designed as a BCR-ABL inhibitor for treatment of chronic myeloid leukemia (CML). Dasatinib targets not only BCR-ABL but also cKIT, platelet-derived growth factor receptor (PDGFR), and SRC family kinases (SFKs), and it is currently mainly used for the treatment of CML patients positive for Philadelphia chromosome [65]. Nevertheless, despite dasatinib cannot be considered a proper anti-angiogenic TKIs, based on the constitutive activation of both PDGFR and SRC kinases in MMECs and MM cells, it efficiently reduces tumor growth and angiogenesis in vitro and in vivo studies in MM xenograft model. It also shows a synergistic cytotoxic effect against MM cells in combination with bortezomib and thalidomide [66]. Although dasatinib underwent phase I/II studies for the treatment of relapsed MM patients as a single agent or in combination (NCT00560352, NCT00429949, NCT01116128, NCT00560391), it is not currently used as standard care for the management of MM patients.

Interestingly, although some multikinase inhibitors do not directly target angiogenic receptors (i.e., VEGFR), they have an anti-angiogenic effect suggesting that they may act as bystander anti-angiogenic drugs.

Overall, these studies demonstrate that despite the promising preclinical results, inhibition of tyrosine kinases is not significantly effective in MM patients. These data suggest that the different basal activation of tyrosine kinases might strongly influence response to therapy and that other factors may be involved in BM angiogenesis and/or resistance to anti-angiogenic therapy in vivo.

### 3.2. Antimyeloma Drugs with Secondary Anti-Angiogenic Properties

#### 3.2.1. Immunomodulatory Drugs

Thalidomide is the first-in-class IMiDs initially used as a hypnotic sedative and subsequently withdrawn because of teratogenic effects [67]. In 2009, thalidomide received the designation as orphan drug for the treatment of relapsed and refractory MM patients by the European Medicines Agency (EMA). More recently, new thalidomide-analogs with lower side effects have been synthetized and approved for clinical practice, i.e., lenalidomide and pomalidomide [68]. IMiDs have pleiotropic anti-tumor activities. Indeed, they enhance an immune anti-tumor response, but also exert a cytotoxic activity against MM cells and inhibit tumor-associated angiogenesis [69] (Figure 2d). The mechanism of action of IMiDs is still unclear. Ito and collaborators [70] identified the ubiquitin E3 ligase cereblon as a teratogenic target of thalidomide. Inhibition of cereblon expression in human MM cell lines significantly reduces cell growth and viability, and correlates to lenalidomide resistance [71], suggesting a key role of this ligase in mediating the IMiDs activity. However, other pathways are likely involved in the IMiDs activity, potentiating their antimyeloma and anti-angiogenic effects, i.e., the STAT3/SP4, ceramide/sphingosine-1-phosphate, and the Slit2/Robo1 axis [72,73,74]. Although IMiDs have a very similar chemical structure, they display a different pharmacological profile including metabolism, half-life, and side effects. Thalidomide metabolism is not dependent on cytochrome P450 but involves different non-enzymatic hydrolytic reactions that generate up to 50 active metabolites [75]. Lenalidomide is mainly excreted in the urine without being metabolized. In contrast, pomalidomide is mainly processed by CYP1A2 and, to a lesser extent, by other isoforms (CYP2C19, CYP2D6, CYP3A4), producing several pharmacological interactions [76]. Analysis of thalidomide metabolites showed that the 5′-OH-thalidomide has an anti-angiogenic activity in the rat aortic ring assay [77]. Furthermore, thalidomide inhibits the kinase IKKα, thereby preventing NF-κB from reaching the nucleus and binding to DNA, thus reducing the production of cell adhesion molecules and inflammatory/angiogenic cytokines [78,79]. In vitro treatment of MMECs with thalidomide significantly reduces the expression and release of angiogenic cytokines, such as VEGF, FGF-2, and HGF [80], implying a modulation of cytokine/growth factors in the BM milieu of thalidomide-treated MM patients. Preclinical studies on lenalidomide confirmed the anti-angiogenic activity of IMiDs. Indeed, lenalidomide reduces the MMECs migration, chemotaxis, and angiogenesis in vitro and in vivo in the CAM assay via the inhibition of the VEGF/VEGFR2 signaling. Lenalidomide also downregulates key genes involved in the MM angiogenesis (*VEGF* and *FGF-2*) and cell migration (*CCL2* and *CXCL12*), and other genes governing the angiogenic process (*BNIP3*, *IER3*, *SEPW1*, *COL4A1*, *CRYAB*, and *Notch1*). This activity implies that the drug exerts multiple effects by acting on different genes and pathways [81].

Overall, IMiDs have shown an anti-angiogenic activity independent from their immunomodulatory effect in vitro [82] (Figure 2d). However, clinical data on IMiDs and BM angiogenesis in MM patients are controversial. Some authors showed that BM MVD is significantly reduced in responder patients treated with thalidomide alone or in combination with dexamethasone [83,84], without affecting serum levels of angiogenic cytokines [84]. By contrast, Cury and collaborators [85] did not find any decrease in BM angiogenesis in patients treated with thalidomide [85]. Similarly, immunohistochemistry analysis of patients’ BM biopsies after treatment with different anti-MM drugs including thalidomide and lenalidomide did not display any significant difference in MVD between responders and non-responders [86]. Despite the uncertain in vivo anti-angiogenic effect, IMiDs represent standard of care for the treatment of MM patients, which have significantly improved their outcome [69].

#### 3.2.2. Proteasome Inhibitors

These drugs are very effective for the treatment of MM patients. They specifically target the ubiquitin proteasome pathway leading to the accumulation of unfolded and misfolded proteins (i.e., immunoglobulins in MM cells) that results in cell death due to accumulation of reactive oxygen species (ROS), endoplasmic reticulum stress, and p53 activation [87].

Bortezomib was the first-in-class proteasome inhibitor approved for the treatment of MM. It specifically targets the chymotrypsin-like site of the 20S subunit. It is available for subcutaneous and intravenous administration with a rapid distribution phase and a longer elimination phase. Since it is extensively metabolized by hepatic cytochrome P450 (CYP3A4 and CYP2C19 isoforms), its pharmacokinetic profile could be strongly influenced by inducers and/or inhibitors of these enzymes [88]. Thrombocytopenia, fatigue, neuropathy, and neutropenia are the most common grade 3 adverse events. More recently, novel proteasome inhibitors with an improved safety profile and ability to overcome bortezomib resistance have been synthetized: carfilzomib, ixazomib, marizomib, oprozomib, and delanzomib [89].

Several studies demonstrated the anti-angiogenic effects of proteasome inhibitors highlighting the ability of these drugs to affect the BM microenvironment [90,91,92,93,94,95] (Figure 2e). Tamura and collaborators [90] showed that bortezomib is cytotoxic against ECs through the modulation of G2/M cell cycle regulators, which blocks cell cycle progression at the M phase and induces apoptosis. This potent anti-angiogenic effect results in increased vascular permeability, suggesting a substantial disturbance of ECs and blood vessel function [90]. Furthermore, treatment of ECs with bortezomib induces an early autophagic response characterized by the accumulation of ROS and LC3-II-positive autophagosomes [91,92]. Inhibition of autophagy through 3-MA or hydroxychloroquine decreases bortezomib-induced cell death, implying the involvement of both autophagy and apoptotic pathways in the ECs response to bortezomib [91,92]. Interestingly, the anti-angiogenic effect of bortezomib is stronger against tumor-associated ECs. Indeed, the pro-apoptotic effect of bortezomib is increased in FGF-activated ECs compared to quiescent ones [91], as well as in ECs cultured in hypoxia-mimicking conditions [93]. Bortezomib inhibits multiple angiogenic capabilities of MMECs including migration, chemotaxis, and adhesion to fibronectin, in vitro (Matrigel^®^), and in vivo (CAM assay) angiogenesis through direct and indirect mechanisms. Bortezomib-treated MMECs display a reduced expression of angiogenic genes, i.e., *VEGF*, *IL-6*, *IGF-I*, and *angiopoietin-1* and *-2* [94]. Furthermore, bortezomib treatment decreases the proliferation of MMECs cultured alone or in the presence of MM cells abrogating the protective effect of tumor cells [94]. Analysis of BM biopsies and sera from bortezomib-treated MM patients showed a significant reduction of CD34^+^ ECs and angiogenin levels confirming its potent anti-angiogenic activity in vivo [95].

The clinical efficacy of bortezomib has been tested in different trials alone and in combination with several antimyeloma drugs, i.e., dexamethasone, thalidomide, lenalidomide, and daratumumab [89]. Overall, bortezomib has great efficacy for the treatment of MM, achieving a deep response and improving the patient’s progression-free survival and quality of life. To date, several clinical trials are testing the safety and efficacy profile of carfilzomib and ixazomib in many combined regimens with promising antimyeloma activity [89].

#### 3.2.3. Bisphosphonates

Bone disease is a common feature of MM that negatively affects patients’ outcomes and quality of life [96]. To counteract and prevent bone resorption and osteoclast activity, patients are treated with bisphosphonates. Bisphosphonates are small orally available compounds poorly absorbed by the gastrointestinal tract and excreted unmetabolized in urine [97]. They have a strong affinity to bone hydroxyapatite crystals, where they are retained for a long time. Here, osteoclasts internalize bisphosphonates that, in turn, inhibit the intracellular farnesyl pyrophosphate synthase, leading to osteoclast apoptosis and preventing bone resorption [97]. Bisphosphonates have been tested for the treatment of MM in combination with other antimyeloma drugs [96]. Analysis of clinical studies showed that bisphosphonates significantly reduce vertebral fractures and bone pain. Nevertheless, long-term exposure to bisphosphonates was associated with an increased risk of jaw osteonecrosis and kidney failure [98]. Zoledronic acid also has an in vitro anti-tumor activity against MM cell lines alone and in combination with dexamethasone and thalidomide [99,100]. Scavelli and collaborators [101] demonstrated that zoledronic acid has a direct anti-angiogenic effect by reducing proliferation, chemotaxis, in vitro (Matrigel^®^), and in vivo (CAM assay) angiogenesis [101] (Figure 2f). These anti-angiogenic effects are enhanced by the reduction of VEGF and VEGFR2 expression, which results in a disruption of the VEGF-mediated autocrine loop [101]. Furthermore, zoledronic acid treatment increases disease-free survival in vivo in the 5T2MM mice model [102]. Accordingly, it prevents bone disease by reducing osteolytic lesions and bone loss and increasing bone mineral density. Zoledronic acid also exerts an antimyeloma activity reducing M protein levels, tumor burden, and vascularization. Immunohistochemistry analysis of CD31^+^ ECs in the tibias and femora from 5T2MM mice treated for short-term with zoledronic acid displays a decreased MVD [102]. Moschetta and collaborators [103] demonstrated the synergistic effect of zoledronic acid and bortezomib in reducing the angiogenic and vasculogenic activities of macrophages from MM patients. Zoledronic acid alone or in combination with bortezomib decreases macrophage proliferation, adhesion, and migration, as well as their ability to form capillary-like structures in capillarogenesis assay in vitro. Finally, zoledronic acid-treated macrophages release lower levels of angiogenic cytokines (i.e., VEGF, bFGF, HGF, and PDGF) and down-regulate the expression of vasculogenic markers, i.e., FVIII-RA, VEGFR2, Tie2/Tek, and VE-cadherin suggesting that this strategy may be useful to counteract vasculogenesis mimicry mediated by MM macrophages [103] (Figure 2f).

#### 3.2.4. Alkylating Agents

These were the first drugs effective for the treatment of MM, and, to date, they still represent one of the most important standards of care for the management of MM patients [104]. Alkylating agents add an alkyl residue to DNA (mainly to N7 of guanine) that results in the excision of guanine, mutations, and/or formation of intra- and inter-strand cross-linking, leading to DNA fragmentation, inhibition of nucleic acids synthesis, and, ultimately, to death of highly proliferative cells. Alkylation also involves proteins and other cellular components, further enhancing their cytotoxic effect [105].

Melphalan is a widely used alkylating agent for the treatment of MM patients in combination with other drugs, such as bortezomib, prednisone, thalidomide, and daratumumab [104]. It can be administered as intravenous or as oral therapy with a better bioavailability if administered on an empty stomach [106]. It is actively uptaken by tumor cells through amino acid carriers with biphasic Michaelis–Menten kinetics [107]. Pharmacodynamic analysis of melphalan in MM patients showed that higher drug exposure is associated with increased toxicity and efficacy mainly due to its plasma proteins-unbound form [108]. The main toxicities are cytopenia, and gastrointestinal toxicity, including anorexia, mucositis, nausea, vomiting, and diarrhea [108]. Some authors demonstrated an anti-angiogenic effect of melphalan in vitro and in vivo in retinoblastoma xenografted mice [109], suggesting that the anti-angiogenic activity may be mediated by NF-κB1 and MYC and their target genes [110].

Melphalan-flufenamide (mel-flufen) is a newly developed alkylating agent [111]. It is a pro-drug of melphalan activated by the aminopeptidases overexpressed by tumor cells, going on to a faster and greater intracellular drug accumulation [111]. Aminopeptidases are overexpressed in MM cells and associated with disease progression and sensitivity to mel-flufen [112]. Mel-flufen shows a higher anti-tumor activity, both in vitro and in vivo (xenografted mice), compared to melphalan, by triggering cell apoptosis via caspases and PARP activation, ROS formation, cytochrome c release, and induction of DNA damage [113]. Mel-flufen inhibits in vitro angiogenesis by significantly reducing the number of vessel branching points and arborization. Furthermore, it also reduces VEGF-induced migration of MM cells through transwell insert without affecting their viability, suggesting that cell homing and metastatization are halted [113]. Gebraad and collaborators [114] showed that BMSCs are sensitive to mel-flufen, abrogating their protective effect on MM cells and affecting BMSCs-mediated angiogenesis. Accordingly, mel-flufen inhibits angiogenesis of HUVEC co-cultured with BMSCs through the reduction of ECs sproutings and of α-SMA-positive pericytes [114] (Figure 2g). Mel-flufen, in combination with dexamethasone, significantly improved MM patients’ overall survival and was approved for relapsed/refractory patients resistant to proteasome inhibitors, IMiDs, and anti-CD38 mAbs (NCT02963493). Other phase I/II clinical trials are studying the safety and efficacy profile of mel-flufen in combination with daratumumab/dexamethasone (NCT04649060) and bortezomib/dexamethasone (NCT03481556) or pomalidomide (NCT03151811).

#### 3.2.5. Glucocorticoids

Dexamethasone and prednisone are analogs of steroid hormones used for the treatment of MM patients for over 50 years [115]. Glucocorticoids bind to their cytosolic receptors, which, after ligand engagement, translocate to the nucleus and modulate the expression of different genes involved in the regulation of several metabolic pathways (gluconeogenesis, glycogenolysis, and lipolysis) and of the anti-inflammatory and immunosuppressive response [116]. The most common adverse events of glucocorticoid treatment in MM patients are the increased risk for hyperglycemia, infections, thrombosis, and neuropsychiatric disorders [117]. Glucocorticoids have a potent antimyeloma effect as a single agent as well as in combination in various clinical trials [115]. Dexamethasone activates the intrinsic pro-apoptotic pathway in different MM cell lines through the upregulation of pro-apoptotic genes and the down-regulation of anti-apoptotic ones leading to MM cell death [118,119].

In addition, glucocorticoids have an anti-angiogenic effect in different solid tumors by decreasing ECs proliferation, migration, and sprouting and by reducing the release of pro-angiogenic cytokines [120]. In MM, treatment of RPMI8226 cells and HUVEC with prednisolone reduces cell viability and induces cell apoptosis indicating its potential efficacy not only against tumor cells but also against blood vessels. Prednisolone also decreases the production of pro-angiogenic cytokines, including VEGF, FGF-2, MMP-2, and MMP-9, and the activation of the angiogenesis-related pathway, such as Notch-1 and NF-κBp65, suggesting that it displays a direct and indirect anti-angiogenic effect by inhibiting the Notch1/NF-Kb axis and by modulating the angiogenic milieu [121] (Figure 2h).

## 4. Concluding Remarks

The “angiogenic switch” is a crucial feature of MM progression and patients’ prognosis. MM cells, as well as BMSCs, contribute to creating a pro-angiogenic niche leading to enhanced angiogenesis and MM progression [5,6,7,8,9]. Different standard-of-care anti-MM drugs have an anti-angiogenic effect, in vitro and/or in vivo, providing promising data on the importance of targeting BM angiogenesis to potentiate the anti-tumor activity of chemotherapeutic agents in MM settings.

Other anti-MM drugs, actually used in the clinical practice, have not shown any anti-angiogenic activity as a single agent, i.e., the anti-CD38 and anti-SLAMF7 mAbs (daratumumab, elotuzumab) and the conventional drug, doxorubicin, but have shown an appropriate/adequate efficacy in combination regimens with anti-angiogenic drugs [89,122,123,124].

Despite the established pathogenic role of angiogenesis in MM, the direct targeting of angiogenesis did not show the expected results [11]. Indeed, although anti-VEGF therapies (i.e., anti-VEGF or anti-VEGFR2 mAbs) or TKIs have shown promising results in preclinical studies, they failed to show a sizable efficacy in MM patients, suggesting that other factors (i.e., extracellular vesicles, miRNAs, lncRNAs, cytokines mixture/cocktail) may be involved in the angiogenic process and may allow therapy escape and drug resistance [34,35,36,37,44,45]. These unfavorable results have downplayed the role of anti-angiogenesis targeting in cancer therapy and highlighted the importance of combo regimens based on anti-tumor and anti-angiogenic drugs.

## 5. Conclusions

Despite the limited efficacy of anti-angiogenic drugs in MM patients, angiogenic inhibition may still represent an attractive strategy to limit tumor growth and MM progression. Hence, further studies are essential to better elucidate the complex mechanisms occurring in the BM angiogenic niche and to design combined strategies to “fight” BM angiogenesis on multiple fronts.

## Figures and Tables

**Figure 1 cancers-15-01990-f001:**
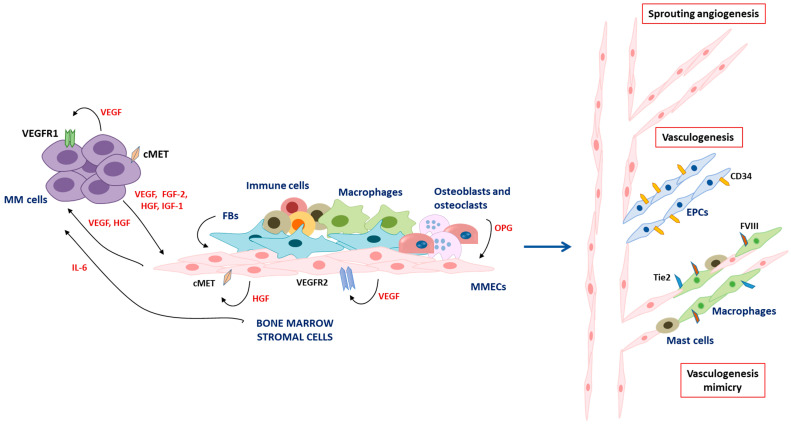
Graphical representation of BM angiogenesis and vasculogenesis in MM patients. MM cells release angiogenic cytokines (i.e., VEGF, FGF-2, HGF, IGF-1) that activate the BM niche. In turn, activated MMECs strongly express angiogenic receptors and secrete high levels of VEGF and HGF that act in an autocrine and paracrine manner. BMSCs contribute to this process by releasing angiogenic cytokines and IL-6, further sustaining the “angiogenic switch” and MM pathogenesis. This pro-angiogenic milieu leads to the formation of new blood vessels via: (i) sprouting angiogenesis; (ii) vasculogenesis through the recruitment of CD34^+^ endothelial progenitor cells (EPCs); and (iii) vasculogenesis mimicry through the recruitment of macrophages and mast cells.

**Figure 2 cancers-15-01990-f002:**
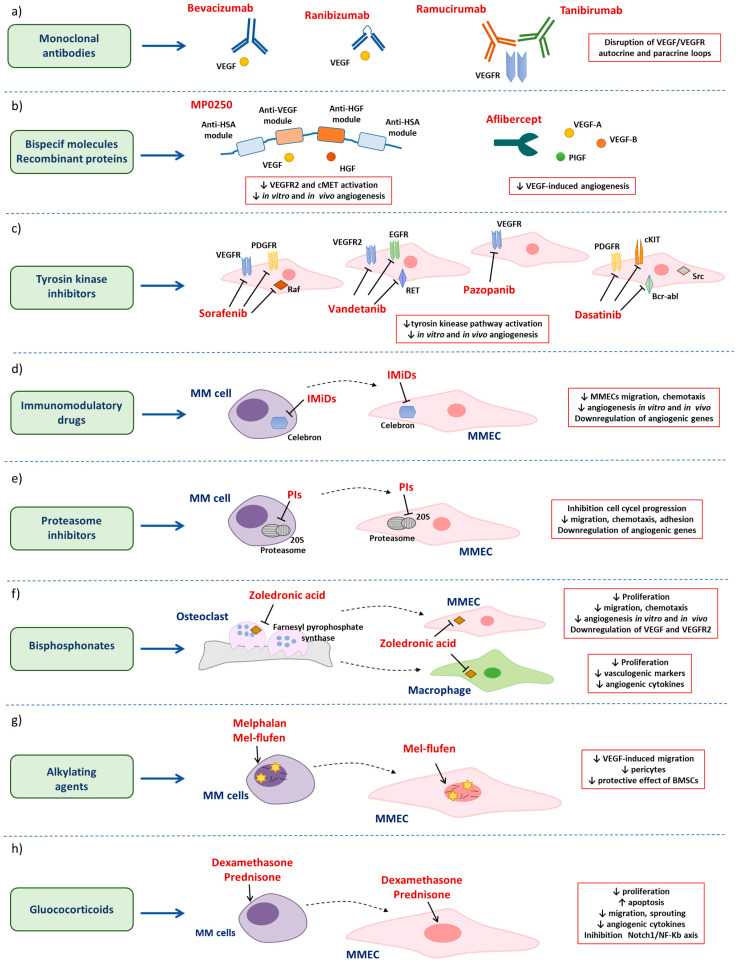
Graphical representation of the main drugs with a direct and indirect anti-angiogenic effect in MM: (**a**) monoclonal antibodies; (**b**) bispecific molecules and recombinant proteins; (**c**) tyrosin kinase inhibitors; (**d**) immunomodulatory drugs; (**e**) proteasome inhibitors; (**f**) bisphosphonates; (**g**) alkylating agents; and (**h**) glucocorticoids. Detailed description in the main text.

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
