# Peer review of "Anti-Angiogenic Activity of Drugs in Multiple Myeloma"

_cancers, 2023, doi:10.3390/cancers15071990_

Round 1

Reviewer 1 Report

This manuscript is a review article on the present understanding of involvement of angiogenesis in progression of multiple myeloma (MM) with a reference survey results of the available anti-angiogenic drugs. It is in general well-written but there are some points which should be reconsidered.

 As mentioned in the text, all the drugs described in section 3 are not anti-angiogenic drugs in their nature. For example, Vandetanib can block VEGFR pathway but originally designed to block EGFR pathway. Other drugs, namely, Immunomodulatory drugs, Proteasome inhibitors, Bisphosphonates, Alkylating agents, and Glucocorticoids should not be categorized as anti-angiogenic drugs. These drugs may have some anti-angiogenic effects but as only secondary or bilateral effects. Therefore, the present form of section 3 is not appropriate because it is liable to lead misunderstanding. The authors should discriminate among 1) proper anti-angiogenic drugs such as Bevacizumab, a humanized anti-VEGF-A IgG1 mAbs, 2) bystander anti-angiogenic drugs such as Vandetanib, and 3) other drugs which show some just secondary anti-angiogenic effects.

 The authors wrote in “Conclusion” that anti-angiogenic drugs failed to show a great efficacy in MM patients, suggesting that other factors (i.e., extracellular vesicles, miRNAs, lncRNAs, cytokines mixture/cocktail) may be involved in the angiogenic process and may allow therapy escape and drug resistance. This conclusion is reasonable, but it also suggests that angiogenesis is just an accessory manifestation of MM and consequently MM therapy should focus on primary tumor cells but not angiogenesis.

Author Response

Author's Reply to the Reviewer 1 Report

Reviewer 1

This manuscript is a review article on the present understanding of involvement of angiogenesis in progression of multiple myeloma (MM) with a reference survey results of the available anti-angiogenic drugs. It is in general well-written but there are some points which should be reconsidered.

The authors thank the Reviewer 1 for helpful criticism and are glad for positive comments.

Reviewer’s comment: As mentioned in the text, all the drugs described in section 3 are not anti-angiogenic drugs in their nature. For example, Vandetanib can block VEGFR pathway but originally designed to block EGFR pathway. Other drugs, namely, Immunomodulatory drugs, Proteasome inhibitors, Bisphosphonates, Alkylating agents, and Glucocorticoids should not be categorized as anti-angiogenic drugs. These drugs may have some anti-angiogenic effects but as only secondary or bilateral effects. Therefore, the present form of section 3 is not appropriate because it is liable to lead misunderstanding. The authors should discriminate among 1) proper anti-angiogenic drugs such as Bevacizumab, a humanized anti-VEGF-A IgG1 mAbs, 2) bystander anti-angiogenic drugs such as Vandetanib, and 3) other drugs which show some just secondary anti-angiogenic effects.

Reply: We thank the Reviewer for this comment that partially overlaps Reviewer 3’s suggestion. Accordingly, we have now included a new 3.1 section namely “Anti-angiogenic drugs in multiple myeloma” and a new 3.2 section entitled “Anti-myeloma drugs with secondary anti-angiogenic properties”.

Regarding Vandetanib, the drug affects multiple targets including EGFR, VEGFR2, RET, Bruton's tyrosine kinase and other cellular kinases (i.e. EPH receptor and Src), leading to multiple effects on tumor cells and endothelial cells. Specifically, Vandetanib treatment significantly reduced tumor growth in different xenograft models (Ryan et al., Br J Cancer 2005) and decreased the VEGF-dependent proliferation of HUVEC (Wedge et al., Cancer Res 2002).

According to Reviewer’s comment, we have now included a sentence in 3.1.3 section, as follow: “Interestingly, despite some multikinase inhibitors do not directly target angiogenic receptors (i.e. VEGFR), they have an anti-angiogenic effect suggesting that they may act as bystander anti-angiogenic drugs.”

Reviewer’s comment: The authors wrote in “Conclusion” that anti-angiogenic drugs failed to show a great efficacy in MM patients, suggesting that other factors (i.e., extracellular vesicles, miRNAs, lncRNAs, cytokines mixture/cocktail) may be involved in the angiogenic process and may allow therapy escape and drug resistance. This conclusion is reasonable, but it also suggests that angiogenesis is just an accessory manifestation of MM and consequently MM therapy should focus on primary tumor cells but not angiogenesis.

Reply: We thank the Reviewer for his/her comment. We are aware of the importance of targeting primary tumor cells and that anti-angiogenesis may only represent an additional weapon against tumor microenvironment. Indeed, most of the anti-angiogenic drugs have been tested in combo regimens with anti-tumor drugs in order to obtain a greater anti-tumor efficacy. Nevertheless, according to Reviewer’s comment on “Concluding remarks” section, we have included a sentence to highlight the importance to directly target tumor MM cells and we have resized the role of anti-angiogenesis, as follow: “Nevertheless, Despite the widely established pathogenic role of angiogenesis in MM, the direct targeting of angiogenesis did not show the expected results [11]. Indeed, although anti-VEGF therapies (i.e., anti-VEGF or anti-VEGFR2 mAbs) or TKIs have shown promising results in preclinical studies, they failed to show a sizable efficacy in MM patients, suggesting that other factors (i.e., extracellular vesicles, miRNAs, lncRNAs, cytokines mixture/cocktail) may be involved in the angiogenic process and may allow therapy escape and drug resistance [34-37, 44,45]. These unfavorable results have downplayed the role of anti-angiogenesis targeting in cancer therapy and highlighted the importance of combo regimens based on anti-tumor and anti-angiogenic drugs.

In conclusion Nevertheless, angiogenic inhibition represents may still represent an attractive strategy able to reduce limit tumor growth and MM progression, and further studies are essential to better elucidate the complex mechanisms occurring in the BM angiogenic niche and to design combined strategies to “fight” BM angiogenesis on multiple fronts.”

Reviewer 2 Report

Congratulations on this manuscript. 

This is a well-written thorough evaluation of available data on antiangiogenic basic science and clinical treatment of multiple myeloma. The authors made an excellent effort to tackle this complex topic. 

The introduction is appropriate. The body of the manuscript goes over basic science background and then moves to the available treatments that could affect angiogenesis. 

Minor comments:

- in line 152: the sentence reads that bevacizumab is approved for myeloma. If that is what the authors meant, please add the approving body (I do not think it is the FDA). On the other hand, if the authors suggested that the combinations are approved, please simplify the sentence. 

- in line 178: "a5nkyrin" is a typo.

Author Response

Reviewer 2

Congratulations on this manuscript.

This is a well-written thorough evaluation of available data on antiangiogenic basic science and clinical treatment of multiple myeloma. The authors made an excellent effort to tackle this complex topic.

The introduction is appropriate. The body of the manuscript goes over basic science background and then moves to the available treatments that could affect angiogenesis.

The authors thank Reviewer 2 for helpful criticism and are very glad for hi/her positive comments

Reviewer’s comment: in line 152: the sentence reads that bevacizumab is approved for myeloma. If that is what the authors meant, please add the approving body (I do not think it is the FDA). On the other hand, if the authors suggested that the combinations are approved, please simplify the sentence.

Reply: We thank the Reviewer for this suggestion. FDA has approved the use of Bevacizumab alone or in combination for the treatment of different cancers, but not for multiple myeloma. Nevertheless, some clinical trials have tested and are currently evaluating the efficacy of Bevacizumab in multiple myeloma patients. We agree that the term “approved” in line 152 could be misleading. Accordingly, we have modified the sentence as follow: “In MM, different clinical trials of bevacizumab alone (NCT00482495) or in combination with other anti-myeloma drugs including bortezomib (NCT00473590, NCT00464178), thalidomide (NCT00022607), lenalidomide and dexamethasone (NCT00410605) have been investigated and/or are under investigation for the treatment of relapsed or refractory patients.”

Reviewer’s comment: in line 178: "a5nkyrin" is a typo.

Reply: We have corrected the typo.

Reviewer 3 Report

I would be supportive of this article. It is very well written and very informative. The whole topic has been researched extensively and presented well including excellent figures.

“The “angiogenic switch” drives monoclonal gammopathy of undetermined significance (MGUS) to multiple myeloma (MM) transition sustaining disease progression and drug resistance.”

The authors need to play down the role of angiogenesis in myeloma in this sentence. There is a difference between correlation and causation. Sure there is a definite change in angiogenesis in different stages of disease evolution but this does not mean that angiogenesis causes the progression – some other biological process could be causing the progression and then angiogenesis could be a consequence of that. There are 1000s of changes in the myeloma clone and microenvironment and its fairly impossible to workout the relevant importance of these and sequence of events that drive them. Agree it is associated with the transition from MGUS to myeloma and is clearly important in providing a microenvironment that sustains the tumour. The fact that direct targeting of angiogenesis has proved disappointing means there needs to be some caution as we clearly do not understand its role as yet.

Many anti-angiogenic drugs have been developed and introduced for the management of myeloma

I think this sentence is misleading as it implies that these drugs were used for their anti-angiogenic properties which is not the case.  I think you could say that many of the drugs used in myeloma have interesting antiangiogenic properties and in this review you will discuss these.

Section 3: Anti-angiogenic drugs in MM

There follows a long list of drug classes. It would be much better to divide this in two 1) Established anti-myeloma drugs with anti-angiogenic properties (immunomodulatory drugs, proteasome inhibitors, bisphosphonates, alkylating agents, glucocorticoids 2) Experimental anti-angiogenic drugs which so far have proved disappointing

Author Response

Reviewer 3

I would be supportive of this article. It is very well written and very informative. The whole topic has been researched extensively and presented well including excellent figures.

The authors thank the Reviewer 3 for helpful criticism and are glad for his/her positive comments.

Reviewer’s comment: “The “angiogenic switch” drives monoclonal gammopathy of undetermined significance (MGUS) to multiple myeloma (MM) transition sustaining disease progression and drug resistance.”

The authors need to play down the role of angiogenesis in myeloma in this sentence. There is a difference between correlation and causation. Sure there is a definite change in angiogenesis in different stages of disease evolution but this does not mean that angiogenesis causes the progression – some other biological process could be causing the progression and then angiogenesis could be a consequence of that. There are 1000s of changes in the myeloma clone and microenvironment and its fairly impossible to workout the relevant importance of these and sequence of events that drive them. Agree it is associated with the transition from MGUS to myeloma and is clearly important in providing a microenvironment that sustains the tumour. The fact that direct targeting of angiogenesis has proved disappointing means there needs to be some caution as we clearly do not understand its role as yet.

Reply: We are grateful for Reviewer’s comment. Hence, according to his/her suggestion, we have modified some sentences in the simple summary and in the introduction as follow:

Simple summary: “The “angiogenic switch” contributes to transition from monoclonal gammopathy of undetermined significance (MGUS) to multiple myeloma (MM) sustaining disease progression and drug resistance.”

Introduction: “A hallmark of this process is the “angiogenic switch” characterized by the formation of new blood vessels. Enhanced angiogenesis, together with other factors (i.e. cytokines, extracellular vesicles, immune escape, ncRNA), fosters MM progression and drug resistance [5].”

Reviewer’s comment: Many anti-angiogenic drugs have been developed and introduced for the management of myeloma.

I think this sentence is misleading as it implies that these drugs were used for their anti-angiogenic properties which is not the case. I think you could say that many of the drugs used in myeloma have interesting antiangiogenic properties and in this review you will discuss these.

Reply: We thank the Reviewer for this suggestion. We agree that this sentence could be misleading and we have modified it as follow: “Based on the pivotal role of angiogenesis in MM progression and its impact on patients’ prognosis, anti-angiogenesis therapy represents an attractive tool for the treatment of MM patients [10,11]. Furthermore, many anti-myeloma drugs have shown secondary anti-angiogenic properties in vitro and in vivo, suggesting a promising potential of angiogenesis targeting. In this review, we describe the most important drugs with a direct and indirect anti-angiogenic effect used in MM settings.”

Reviewer’s comment: Section 3: Anti-angiogenic drugs in MM. There follows a long list of drug classes. It would be much better to divide this in two 1) Established anti-myeloma drugs with anti-angiogenic properties (immunomodulatory drugs, proteasome inhibitors, bisphosphonates, alkylating agents, glucocorticoids 2) Experimental anti-angiogenic drugs which so far have proved disappointing

Reply: We thank the Reviewer for this comment that partially overlaps Reviewer 1’s suggestion. Accordingly, we have now included a new 3.1 section namely “Anti-angiogenic drugs in multiple myeloma” and a new 3.2 section entitled “Anti-myeloma drugs with secondary anti-angiogenic properties”.

Round 2

Reviewer 1 Report

The manuscript has been revised in response to the reviewer’s comments and refined considerably. One minor revision is desirable. There is a sense of discomfort in describing dasatinib as an anti-angiogenic drug. It is originally designed as a BCR-ABL inhibitor for treatment of CML and different from the other tyrosine kinase inhibitors in section 3.1.  So, some annotation is needed if it is included in anti-angiogenic drugs.

Author Response

The authors thank the Reviewer 1 for helpful criticism and are glad for positive comments.

Reviewer’s comment: The manuscript has been revised in response to the reviewer’s comments and refined considerably. One minor revision is desirable. There is a sense of discomfort in describing dasatinib as an anti-angiogenic drug. It is originally designed as a BCR-ABL inhibitor for treatment of CML and different from the other tyrosine kinase inhibitors in section 3.1.  So, some annotation is needed if it is included in anti-angiogenic drugs.

Reply: We thank the Reviewer for this comment. Accordingly, we have revised the paragraph describing Dasatinib as follow: “Finally, dasatinib is a TKI originally designed as a BCR-ABL inhibitor for treatment of chronic myeloid leukemia (CML). Dasatinib targets not only BCR-ABL but also cKIT, platelet-derived growth factor receptor (PDGFR), and SRC family kinases (SFKs), and it is currently mainly used for the treatment of CML patients positive for Philadelphia chromosome [65]. Nevertheless, despite Dasatinib cannot be considered a proper anti-angiogenic TKIs, based on the constitutive activation of both PDGFR and SRC kinases in MMECs and MM cells, it efficiently reduces tumor growth and angiogenesis in vitro and in vivo studies in MM xenograft model. It also shows a synergistic cytotoxic effect against MM cells in combination with bortezomib and thalidomide [66]. Although dasatinib underwent phase I/II studies for the treatment of relapsed MM patients as a single agent or in combination (NCT00560352, NCT00429949, NCT01116128, NCT00560391), it is not currently used as standard care for the management of MM patients.”
